# Quorum Quenching Enzyme (PF-1240) Capable to Degrade AHLs as a Candidate for Inhibiting Quorum Sensing in Food Spoilage Bacterium *Hafnia alvei*

**DOI:** 10.3390/foods10112700

**Published:** 2021-11-05

**Authors:** Yue Shen, Fangchao Cui, Dangfeng Wang, Tingting Li, Jianrong Li

**Affiliations:** 1National & Local Joint Engineering Research Center of Storage, Processing and Safety Control Technology for Fresh Agricultural and Aquatic Products, College of Food Science and Technology, Bohai University, Jinzhou 121013, China; sy09282021@163.com (Y.S.); cfc1031@163.com (F.C.); 2School of Food Science and Technology, Jiangnan University, Wuxi 214000, China; 15941611651@163.com; 3Key Laboratory of Biotechnology and Bioresources Utilization, Dalian Minzu University, Dalian 116000, China

**Keywords:** quorum sensing quenching enzyme, *Hafnia alvei*, biofilm, extracellular protease, motility

## Abstract

Quorum sensing (QS) is widely present in microorganisms in marine aquatic products. Owing to the use of antibiotics, many spoilage bacteria in aquatic products are drug resistant. In order to slow down this evolutionary trend, the inhibition of spoilage phenotype of spoilage bacteria by interfering with QS has become a research hot spot in recent years. In this study, we found a new QS quenching enzyme, PF-1240; it was cloned and expressed in *Pseudomonas fluorescens* 08. Sequence alignment showed that its similarity with *N*-homoserine lactone (AHL) acylase QuiP protein of *Pseudomonas fluorescens* (*Pf* 0-1) was 78.4%. SDS-PAGE confirmed that the protein is a dimer composed of two subunits, which is similar to the structure of AHL acylases. The concentration of heterologous expression in *Escherichia coli* (DE3) was 26.64 μg/mL. Unlike most AHL acylases, PF-1240 can quench AHLs with different carbon chain lengths and inhibit the quorum sensing of the aquatic spoilage bacterium *Hafnia alvei*. It can significantly reduce the formation rate of biofilm of *H. alvei* to 44.4% and the yield of siderophores to 54%, inhibit the production of protease and lipase, and interfere with the motility of *H. alvei*. Through these corruption phenotypes, the specific application effect of PF-1240 can be further determined to provide a theoretical basis for its application in the preservation of practical aquatic products.

## 1. Introduction

Quorum sensing (QS) is a method of information exchange between bacteria. Both G^+^ and G^-^ bacteria use this chemical-based form of intercellular communication to regulate a variety of physiological activities [1]. This can achieve the purpose of communication by secreting signal molecules in order to regulate the expression of corresponding functional genes and make bacteria show some phenotypic characteristics such as the secretion of virulence factors, formation of biofilm, and enhancement of bacterial motility and diffusion [2,3].

In recent years, many researchers have focused on the study of QS in aquaculture [4], health care [5], phytoremediation [6], and wastewater treatment [7]. Traditional antibiotics can kill bacteria or inhibit their growth, but with the use of a large number of antibiotics, drug resistance of pathogenic bacteria has appeared. QS enables bacteria to regulate their collective behavior depending on the cell density of their environment. Therefore, we can reduce the risk of pathogens by interfering with QS [8]. The pathogenicity of pathogenic bacteria in aquaculture is regulated by QS, and the virulence factors and biofilm produced lead to the infection of aquatic products, thus causing adverse effects on the public health of the environment [9].

Owing to the existence of microorganisms in the process of aquaculture and the infection of microorganisms in the process of transportation, the main reason for the deterioration of fresh aquatic products is the growth and metabolism of microorganisms. QS systems exist in many aquatic spoilage bacteria, which can participate in and regulate the spoilage process of aquatic products such as the production of extracellular protease (EPS), biofilm formation, and mucus secretion. This has a serious impact on the quality of aquatic products. Most spoilage bacteria are G^−^ bacteria. They use AHLs as signaling molecules for QS [10]. Several bacteria related to food spoilage produce AHL signals, including members of the genera *Pseudomonas*, *Serratia*, and *Hafnia* [11].

*H. alvei* is a common dominant spoilage bacterium in vacuum packaging and modified atmosphere packaging food, and it is also an opportunistic pathogen [12]. It has been widely separated from foods stored at low temperatures such as raw meat, dairy products, and aquatic products [13]. The results showed that *H. alvei* is the dominant spoilage bacterium in many aquatic products [14] such as salmon and turbot [15]. It can produce a variety of AHLs and mediate the expression of various spoilage factors through QS [16].

Suppressing the spoilage of aquatic products by interfering with QS can not only reduce the selective pressure of bacteria but also effectively reduces the drug resistance caused by the overuse of antibiotics [17]. Quorum quenching (QQ) destroys the bacterial QS system by inhibiting the synthesis and accumulation of signal molecules or by degrading and modifying signal molecules, thus blocking the adverse effects of spoilage bacteria or pathogenic bacteria [18]. QQ enzymes can use AHLs as substrates to decompose AHLs through enzymatic reactions. 

According to the different catalytic mechanisms of quenching enzymes, they can be divided into lactonase, acylase, and oxidoreductase. AHL acylases are well-known members of the Ntn-hydrolase family of proteins that inactivate AHLs by cleaving the acyl side chains from the homoserine lactone [19]. The biochemical properties of these enzymes have been studied, including their kinetic properties, stability, and ability to control microorganisms in vitro and in vivo for a variety of applications from medical devices to animal health and agriculture [20,21]. 

Compared with AHL lactonase, AHL acylase has more advantages in practical applications. This is because compared with the degradation product *N*-acyl homoserine (which can be recycled to AHL at acidic pH), the acylase reaction product cannot spontaneously regenerate active functional AHLs. In addition, fatty acids produced by acylases are usually easily metabolized [22]. However, at present, most AHL acylase substrates are not broad-spectrum, and PvdQ, which is the most studied, has an irreversible degradation effect on long-chain AHL [23]. In 2014, Ruchira et al. [24] reported for the first time the activity of penicillin G acylase (KcPGA) AHL acylase from *Kluyvera citrophila*. Biochemical analysis and molecular docking showed that KcPGA had activity on AHLs with six to eight carbon atoms regardless of whether it was modified by the 3-oxy group.

Most QS bacteria produce diverse AHLs, so the development of QQ enzymes with broad-spectrum quenching activity has important application value. In this study, we identified a gene encoding a potential AHL acylase named PF-1240 from the genome of *P. fluorescens* 08 isolated from turbot. Turbot is native to the northeast coast of the Atlantic Ocean. It is called “Duobao fish” in China. It is a cold water fish with great economic value [25]. The protein sequence of PF-1240 was 78.4% identical to that of the QuiP protein. It has been found that the QuiP gene (PA1032 gene) of *Pseudomonas aeruginosa* can encode AHL acylase and degrade the 3O-oxo-C_12_-HSL molecule, which promotes the bacteria to enter a static state and reduces energy consumption [26]. Therefore, it is speculated that the PF-1240 protein of *P. fluorescens* may also have the ability to degrade AHLs. We successfully expressed the PF-1240 protein by genetic engineering and analyzed its enzymatic properties. Its inhibitory effects on the putrefaction phenotype of *H. alvei* was determined and include biofilm formation; the production of EPS, siderophores, proteases, and lipases; swarming; and swimming. This highlights the potential of PF-1240 as a novel QQ enzyme.

## 2. Methods and Materials

### 2.1. Bacterial Strains, Growth Conditions, and Reagents

A spoilage bacterium, *H. alvei*, which produces multiple AHLs, was isolated from a spoiled turbot. *Chromobacterium violaceum* CV026 was used to detect short-chain AHLs. *E. coli BL21* (DE3) cells were used to express the target protein. The three strains were grown in LB medium. *C. violaceum* CV026 was supplemented with 20 μg/mL kanamycin. *E. coli BL21* (DE3) cells were supplemented with 50 μg/mL kanamycin. Isopropyl-β-D-thiogalactopyranoside (IPTG), kanamycin, X-gal, gentamicin, spectinomycin, and glutaraldehyde were purchased from Beijing Solarbio Science Technology Co., Ltd. (Beijing, China). SDS-PAGE kits, His-Select Nickel Affinity Gel, BCA Protein Assay Kit, and Fast Blue Protein Stain Solution were purchased from Sangon Biotech Co., Ltd. (Shanghai, China). The AHLs used in this study were *N*-butanoyl-DL-homoserine lactone (C_4_-HSL; Sigma, St. Louis, MI, USA), *N*-3-hexanoyl-DL-homoserine lactone (C_6_-HSL; Sigma), *N*-3-octanoyl-DL-homoserine lactone (C_8_-HSL; Sigma), *N*-3-decanoyl-DL-homoserine lactone (C_10_-HSL; Sigma), *N*-3-dodecanoyl-DL-homoserine lactone (C_12_-HSL; Sigma), and *N*-tetradecanoyl-DL-homoserine lactone (C_14_-HSL; Sigma). Chrome-Azurol-S CAS-agar was purchased from NC Pharmculture Co., Ltd. (Beijing, China).

### 2.2. Extraction of H. alvei AHLs

*H. alvei* were cultured for 24 h, and 100 mL of the culture was centrifuged at 10,000 rpm for 10 min. Then, the supernatant was mixed with an equal amount of ethyl acetate [0.1% (*v/v*) glacial acetic acid]. The mixture was shaken adequately for 30 s and remained in the layer. After extraction for 6 h, we poured out the ethyl acetate in the upper layer, added new ethyl acetate [0.1% (*v/v*) glacial acetic acid], and repeated the above operation. This process was repeated three times, and the entire ethyl acetate fraction was mixed. The combined ethyl acetate fractions were evaporated using rotary evaporators (35 °C, 150 rpm) to dryness and redissolved in 1 mL of methyl alcohol. The extracts were stored in sterile microcentrifuge tubes at −20 °C, and it was used to determine the effect of PF-1240 on *H. alvei*.

### 2.3. Expression and Purification of PF-1240

In the previous research, we successfully cloned the target gene and sent the obtained target gene sequence to Takara company (http://takara.biogo.net/, accessed on 13 March 2021) for codon optimization and artificial synthesis. The synthesized gene was amplified by PCR to obtain the target fragment for insertion. *Bam*H and *Xho*I were used as restriction sites to connect the target gene with pET-28a (+). The ligation product was transformed into competent *E. coli BL21* (DE3) cells and named pET28-PF-1240-BL21.The protein was expressed in LB liquid medium at 37 °C overnight at 160 rpm.

To induce the recombinant protein overexpression, 0.4 mM isopropyl-β-D-thiogalactopyranoside (IPTG) was added to 300 mL *E. coli* cultures. The cells were harvested by centrifugation at 7500× *g* for 20 min after 14 h of cultivation at 16 °C, washed with PBS, resuspended in the same buffer, and disrupted using an ultrasonic disintegrator. Cell debris was removed by centrifugation at 7500× *g* for 20 min. The supernatant was applied to His-Select Nickel Affinity Gel, and the His-tagged recombinant protein was purified according to the manufacturer’s instructions. The purified protein was treated with protein loading buffer and heated for 10 min at 98 °C. The sample was subjected to a 12% PAGE gel at 80 V for 30 min and 120 V for 90 min. Protein bands were stained with Fast Blue Protein Stain solution.

### 2.4. Determination of PF-1240 Concentration and Validation of Its Activity

Determination of PF-1240 using a BCA protein assay kit. The enzymatic degradation of short-chain AHLs was measured using the biosensor strain CV026.

### 2.5. Enzymatic Properties of PF-1240

The hydrolytic activity of PF-1240 on AHLs was determined using gas chromatography-mass spectrometry (GC-MS Agilent 7890N/5975, Agilent, Palo Alto, CA, USA). According to Li et al. [27]. All sample injections were done in the split mode (50:1) into an HP-5 MS capillary column (30 m length × 0.25 mm internal diameter × 0.25 µm film thickness) (Agilent, Palo Alto, CA, USA). Helium was used as the carrier gas at a flow rate of 1 mL/min. The GC injector temperature was 200 °C, and the oven temperature was programmed as follows: 150 °C ramped at 10 °C/min to 220 °C and ramped at 5 °C/min to 250 °C, then ramped at 0.5 °C/min to 252.5 °C. Mass spectrometry conditions were as follows: the electron ionization source was set to 70 eV, MS Quad 150 °C, emission current 500 µA, MS Source 230 °C. Data were acquired by either full-scan mode (*m/z* –800) or in selected ion monitoring (SIM) mode (m/z 143). The purified enzyme was mixed with C_4_-HSL, C_6_-HSL, C_8_-HSL, C_10_-HSL, C_12_-HSL, and C_14_-HSL standards to reach a final concentration of 10 mmol/L. After reacting at 30 °C for 3 h, the amount of residual AHLs was determined by GC-MS, and the average value was calculated three times. The highest enzyme activity was defined as 100%, and the relative enzyme activity of PF-1240 on different substrates was calculated. In the same way, the remaining content of C_8_-HSL was determined after incubation for 3 h at 20 °C, 25 °C, 30 °C, 35 °C, 40 °C, 45 °C, and 50 °C. Different pH buffers were prepared: pH 6.0–10.0 (phosphate–potassium hydroxide–PBS buffer). PF-1240 was reacted with C_8_-HSL in different concentrations of pH buffer at a reaction temperature of 30 °C. After 3 h, the amount of remaining C_8_-HSL was determined to determine the optimal reaction pH for PF-1240.

### 2.6. Inhibitory Effect of PF-1240 on Spoilage Characteristics of H. alvei

#### 2.6.1. Effect of PF-1240 on Cell Viability

The enzyme was added to the culture medium of *H. alvei* and cultured at 28 °C until the logarithmic growth stage. The bacteria were collected by centrifugation. After washing twice with 0.1 mol/L PBS (pH 7.4), the bacteria were collected by centrifugation for 5 min. PI (0.1%) staining solution was used for staining at 4 °C for 30 min, and apoptosis was detected by flow cytometry within 1 h. The control group was treated with *H. alvei* without PF-1240.

#### 2.6.2. Effect of PF-1240 on Production of Purpurin in CV026

According to Li et al. [27], after overnight activation, strain CV026 was inoculated into LB broth containing different amounts of PF-1240 (0, 5, 10, 15, 20 and 25 μg/mL) at a ratio of 1:100 (*v/v*), and 200 μL of the crude extract of AHLs prepared from 2.2 was cultured at 160 r/min and 28 °C for 48 h. Then, 300 μL of the culture medium was added to a 1.5 mL centrifuge tube, and 150 mL μL 10% sodium dodecyl sulfate was added and shaken for 10 s. Then, 600 μL *N*-butanol was added and shaken for 5 s. This was centrifuged for 5 min at 10,000 r/min, after which absorption for 200 μL purple supernatant was added to a 96-well plate. The absorbance (OD) was measured at 595 nm using a microplate reader, and the experimental group without the signal molecule was used as a control.

#### 2.6.3. Effect of PF-1240 on Biofilm of *H. alvei*

##### Determination of Biofilm by Enzyme Plate Method

The overnight activated *H. alvei* was mixed with fresh Luria-Bertani (LB) medium at a ratio of 1:100 (*v/v*). First, 1 mL of LB medium was placed into a sterile centrifuge tube, and the final concentrations were 5, 10, 15, 20, and 25 μg/mL PF-1240.The negative control was a PBS solution with an equal volume added. After incubation at 28 °C for 48 h, the cell density was determined and centrifuged. This was cleaned with sterile water three times, dried with sterile air, and fixed for 35 min. Then, 1 mL of 0.1 g/100 mL crystal violet was added to the dye for 15 min (at room temperature), cleaned with sterile water, dissolved in 33% glacial acetic acid solution, and OD_595_ nm was determined using an enzyme reader. The relative formation rate of the biofilm was calculated using Formula (1).
(1)$$\mathrm{Biofilm}\;\mathrm{formation}\;\mathrm{rate}/\%=\frac{\mathrm{OD}595\mathrm{nm}\;\mathrm{experience}\;\mathrm{group}}{\mathrm{OD}595\mathrm{nm}\;\mathrm{control}\;\mathrm{group}}\times 100$$

##### SEM Analysis of Biofilms

To evaluate the disrupting effects of PF-1240 on biofilms, overnight cultures of *H. alvei* were inoculated in 2 mL of fresh LB broth containing sterile stainless steel (SS) type AISI 304 (mechanically polished, No. 4 grade; 1 cm × 1 cm × 1 mm thickness) in LB broth liquid medium treated with 5, 10, 15, 20, and 25 μL/mL PF-1240. The control group did not receive PF-1240 therapy. After incubation at 28 °C for 24 h, the SS slices were rinsed with PBS to remove planktonic cells. For scanning electron microscopy (SEM), biofilms were fixed with 2.5% glutaraldehyde for 24 h at 4 °C and gradually dehydrated using a graded series of ethanol (40%, 70%, 90%, and 100%). The SS slices were placed on a silicon wafer, covered with gold palladium, and subjected to SEM.

#### 2.6.4. Determination of Extracellular Polymers

Raman spectroscopy (RS) was used to measure the EPS content of *H. alvei.* Sample processing was performed as described in Section 2.6.3. After removing the zinc sheet from the biofilm, it was washed with PBS three times. The floating bacteria was washed from the surface, and the result was dried for 30 min for the RS test.

#### 2.6.5. Detection of Siderophore

##### CAS Plate Tests

Siderophore production was detected on solid media using Chrome-Azurol-S CAS-agar [27] with some modifications. To prepare 1 L of blue agar, 60.5 mg of CAS was dissolved in 50 mL of distilled water and mixed with 10 mL iron(III) solution (1 mM FeCl_3_·6H_2_O, 10 mM HCl). This solution was slowly added to 72.9 mg hexadecyltrimethyl ammonium bromide (HDTMA) dissolved in 40 mL of distilled water. The composite dark blue liquid was autoclaved at 121 °C for 15 min, keeping the standby as the CAS assay solution. The autoclave was 30 mL casamino acids (10%) after cooling to 50 °C, stirring with 1 mL CaCl_2_ (1 mM), 20 mL MgSO_4_·7H_2_O (1 mM), 10 mL glucose (20%) as a carbon source, 15 g agar, 30.24 g 1,4-piperazinediethanesulfonic acid (Pipes), and 12 g of a 50% (*w/w*) NaOH solution to increase the pH to the pKa of Pipes (6.8). Then, the cultures were heated to 121 °C and maintained for 15 min by autoclaving. After cooling to 60 °C and stirring the CAS assay solution mentioned above, the solution was slowly added to an edgeways conical flask, with sufficient agitation without the generation of foam. Each plate contained 20 mL of blue agar and was punched using an Oxford cup (autoclaved). These blue agars were used to detect siderophores. PF-1240 was added as follows: 5, 10, 15, 20, and 25 μg/mL PF-1240. An equal volume of PBS was used as the negative control.

##### Detection of Siderophore in Liquid Medium

The quantity of iron carrier produced by bacteria in liquid medium was quantified, and the OD_600_ was determined and centrifuged after a certain period of culture. The supernatant was filtered with 0.22 μm microporous membrane, 100 μL supernatant was mixed with quantitative test solution, and sterile water was used for the same operation as control. After the reaction at room temperature, the OD_600_ values of the sample (AS) and the control (Ar) were recorded. The output of siderophore is calculated using the following Formula (2):(2)$$\mathrm{Relative}\;\mathrm{content}\;\mathrm{of}\;\mathrm{Siderophore}=\frac{\mathrm{Ar}-\mathrm{As}}{\mathrm{Ar}}\times 100\%$$

#### 2.6.6. Motility Inhibition Assays

Swarming and swimming motilities were examined using swarming agar and a swimming agar medium. Briefly, PF-1240 was mixed with swarming agar and swimming agar medium, and the final concentrations of PF-1240 were 5, 10, 15, 20, and 25 μg/mL. In addition, 5 μL bacterial suspension of *H. alvei* was added to the center of the cooled plate. The plates were incubated at 28 °C for 48 h to determine the migration diameter.

#### 2.6.7. Determination of Protease

The method of Vijayaraghavan et al. [28] involves making a milk plate, punching holes with an Oxford cup, and adding *H. alvei* supernatant cultured overnight with PF-1240 at different concentrations (5, 10, 15, 20, and 25 μg/mL). This was cultured at 28 °C for 18–24 h. After casein was hydrolyzed by protease, an obvious hydrolysis circle appeared around the pores. The larger the hydrolysis circle, the higher the extracellular protease activity. The supernatant without immobilized penicillin acylase was used as a negative control.

#### 2.6.8. Determination of Lipase

*H. alvei* bacterial solution cultured overnight with PF-1240 at different concentrations (5, 10, 15, 20, and 25 μg/mL) was centrifuged at 4 °C at 8500 rpm for 5 min. The supernatant was collected and used with 0.22 μM sterile filter membrane filtration and sterilization for standby.

Dissolve the following substances with 200 mL water and sterilize them: 4 g agar powder, 0.4 mL triglyceride, 2 g peptone, 1 g yeast powder, and 2 g NaCl. This was poured into a Petri dish with an Oxford cup. After the plate solidified, 200 μL of the supernatant was added to the wells. The cells were cultured at 28 °C for 48 h, and the size of the transparent circle around the hole was observed.

## 3. Results and Discussion

### 3.1. Recombinant Expression of PF-1240 by E. coli BL21

In the present study, we identified one putative AHL acylase gene (*pf*-1240) in the *P. fluorescens* 08 genome based on homology and domain searches. In an amino acid sequence comparison, PF-1240 was related to Ntn hydrolase family proteins and showed the highest identity to AHL acylase (QuiP) (78.4%) from *Pf* 0-1 among the verified acylases (Appendix A). Since the results suggested that PF-1240 had a potential AHL quenching function, we purified the enzyme from the recombinant and isolated *E. coli BL21* (DE3) with a Ni affinity column. SDS-PAGE analysis showed that PF-1240 was composed of two subunits (one below 65 kDa and one below 25 kDa) corresponding to the predicted molecular weight (80 kDa) of the amino acid sequence (724 amino acids). After purification, a clear band was obtained, as shown in Figure 1A. At present, most AHL acylases are double-stranded, in which the β chain is the large subunit and the α chain is the small subunit. The active site, Ser, is located at the head of the β chain and is conserved in different double-stranded AHL acylases. In addition, most of the AHL acylases identified so far are members of a single superfamily, showing *N*-terminal nucleophilic (Ntn) hydrolase folding [19]. The concentration of PF-1240 was determined using a protein concentration determination kit. The enzyme was extracted from 150 mL of bacterial solution, and the protein concentration was determined after dilution. The concentration of crude enzyme solution was 33.81 mg/mL, and the concentration of purified enzyme was 26.64 μg/mL (Appendix A).

### 3.2. Enzymatic Properties of PF-1240

#### 3.2.1. Optimal Substrate of PF-1240

Most AHL acylases tend to degrade long-chain AHLs (with or without a substituent at the C-3 of the acyl chain). However, Kusada et al. [29] found that MacQ was able to inactivate all tested AHLs ranging from C_6_ to C_14_ carbon chains with or without 3-oxo substitutions. AHLs with different acyl chains were used as substrates to study the substrate specificity of PF-1240. In the reaction system, various substrates were added at a certain concentration; after the reaction, the remaining AHLs were extracted with 1 mL of ethyl acetate, the content of the remaining AHLs was determined by GC-MS, and the optimal substrate enzyme activity was 100%. Relative enzyme activities of the other substrates were determined. The hydrolysis ability of PF-1240 to various substrates is shown in Figure 1B. As shown in Figure 1B, PF-1240, as a new QS quenching enzyme, has a broad spectrum of substrates and can degrade AHLs with different acyl chains. The degradation activity of PF-1240 was the strongest for C_8_-HSL, but the degradation activity was lower for AHLs with carbon chain length > 10, and the relative degradation activity was less than 20%.

#### 3.2.2. Optimum Reaction pH of PF-1240

The optimum pH for fish growth is 6.5–8.5, so the study on the optimum pH of PF-1240 will lay a theoretical foundation for the development and utilization of PF-1240 in the future. The purified PF-1240 enzyme solution was placed in buffer solutions with different pH values, and the activity of AHL acylase was determined at 30 °C. The highest enzyme activity was defined as 100%, and the relative enzyme activity at different pH values was calculated. The results showed that the optimum pH value of PF-1240 was 9.5, and the activity of AHL acylase decreased significantly at pH 6. When the pH was between 7 and 9.5, enzyme activity was relatively stable. The results showed that PF-1240 had similar properties to other AHL acylases and maintained high activity under alkaline conditions.

#### 3.2.3. Optimum Reaction Temperature of PF-1240

The purified PF-1240 enzyme solution was reacted with C_8_-HSL at different temperatures to detect the activity of AHL acylase. The highest enzyme activity was defined as 100%, and the relative enzyme activity of PF-1240 at different temperatures was calculated. The results showed that the optimum temperature for PF-1240 was 30 °C. As shown in Figure 1D, with an increase in temperature, the enzyme activity showed a fluctuating downward trend, and the remaining relative enzyme activity was 78% at 40 °C.

### 3.3. Effect of PF-1240 on the Growth of H. alvei

In recent decades, antibiotic resistance (AMR) has become a global problem that seriously threatens human and animal health [30]. The use of the QQ enzyme is fundamentally different from the use of antibacterial agents: the QQ enzyme is not toxic and has little effect on bacterial growth [31]. Flow cytometry was used to determine the effect of different concentrations of PF-1240 on the growth of *H. alvei*. The four quadrants of UL (upper left), UR (upper right), LL (lower left), and LR (lower right) in Figure 2A represent fragmented and damaged cells, late apoptotic and dead cells, normal cells, and early apoptotic cells, respectively. For bacteria, there is no clear mechanism of apoptosis, so we only focus on dead cells and normal cells. The content of normal cells was 95.7–90.7%, and it was found that different concentrations of enzyme had no significant effect on the growth of *H. alvei*. It did not kill *H. alvei*, indicating that PF-1240 conforms to one of the conditions of the QQ enzyme.

### 3.4. Determination of QS Inhibitory Activity of PF-1240

QS relies on the interaction between diffusion signal molecules and transcription activating proteins, coupling gene expression with cell population density. In G^-^ bacteria, this type of signal molecule is usually AHL, and the structure of its *N*-acyl side chain is different. AHL can induce CV026 to synthesize violacein, which is sensitive to AHLs with *N*-acyl side chains from C_4_ to C_8_ [32]. As shown in Figure 2B, adding the crude enzyme solution to the plate containing CV026 and AHLs of *H. alvei* can inhibit the formation of purple bacteriocin around the enzyme.

### 3.5. Effect of PF-1240 on Production of Purple Bacteriocin CV026

By observing the difference in purple bacteriocin produced by CV026, we can see the degree of interference of different enzyme concentrations on QS. The effects of different concentrations of PF-1240 on the production of CV026 purple bacteriocin can be measured quantitatively using the purified enzyme. As shown in Figure 2C, with an increase in PF-1240 concentration, the yield of purple bacteriocin gradually decreased. It is inferred that PF-1240 can inhibit the production of CV026 purple bacteriocin in the presence of exogenous AHLs by interfering with the QS phenomenon. It can be speculated that PF-1240 can inhibit the information exchange between spoilage bacteria by degrading AHLs, thus reducing the production of spoilage factors and achieving the effect of fresh keeping of aquatic products.

### 3.6. Effect of PF-1240 on the Formation of H. alvei Biofilm

Biofilm is an extracellular group composed of EPS, nucleic acids, proteins, and lipids secreted by bacteria. It has a complex three-dimensional (3D) structure. It can provide structural support for bacterial colonies, enhance the adaptability of bacteria to various environmental conditions, and confer resistance to a variety of antibacterial compounds (including antibiotics) [33,34,35]. Biofilm formation is a common source of pollution in aquatic products. QS was found to be involved in spoilage and biofilm formation in fresh meat products during storage under aerobic conditions [36]. Figure 3A shows the effect of different concentrations of PF-1240 on the formation of *H. alvei* biofilms under SEM. It can be seen that the bacteria are tightly wrapped by extracellular substances without the addition of enzymes, and few single bacteria can be seen. With an increase in enzyme concentration, the number of lamellar or aggregated membranes decreased gradually, and single bacterial aggregates appeared. As shown in Figure 3A(f), the number and size of bacterial aggregates decreased significantly, indicating that PF-1240 could significantly inhibit the formation of *H. alvei* biofilm. Figure 3B shows the quantitative change in biofilm formation, which is similar to the SEM results. With an increase in enzyme concentration, the relative formation rate of the biofilm became increasingly smaller. The biofilm formation rate decreased to 44.4% at an enzyme concentration of 25 μg/mL.

### 3.7. Effect of PF-1240 on the EPS of H. alvei

During the growth of microorganisms, a macromolecular substance called EPS is secreted into the cell wall. EPS is a metabolite of microorganisms and is derived from the decomposition of organic matter in the environment. The composition is very complex and is mainly composed of polysaccharides, proteins, humus, nucleic acids, and lipids [37]. With the development of the food industry in recent years, it has been discovered that extracellular polymers are related to the properties that promote cell adhesion to surfaces, such as from drying or desiccation in bacteria growing in terrestrial habitats or from attack by antimicrobial agents [38]. In addition, EPS can fix heavy metals in the environment, thus causing heavy metal pollution in the host [39]. As shown in Figure 3B(a), when no enzyme was added, there were four obvious Raman peaks. The first peak is 1125 cm^−1^, representing the COC stretching vibration and symmetric glycosidic link [40]; the second peak is at 1347 cm^−1^, representing amide III in protein; the third peak is at 1440–1460 cm^−1^, representing deformation vibration CH_2_ scissoring, which is a characteristic of lipids [41]; and the fourth peak is 1600 cm^−1^, representing aromatic ring stretching sporopollenin and phenylpropionamide [42]. With an increase in PF-1240 concentration, the number of Raman vibration peaks decreased, and the peak height decreased. When the concentration of PF-1240 was 25 μg/mL, the Raman spectrum tended to be flat without an obvious peak, indicating that the addition of PF-1240 can effectively inhibit the formation of EPS in *H. alvei*. EPS is the main component of biofilms. With a decrease in extracellular polymer, the formation of biofilm is also affected, which corresponds to the SEM results.

### 3.8. Effect of PF-1240 on Swarming and Swimming of H. alvei

The swarming and swimming of bacteria is the fastest method of surface migration, and the genes regulating these behaviors are affected by the QS system [43]. As shown in Figure 4A,B, with an increase in enzyme concentration, the migration diameter formed by clustering and swimming became increasingly smaller, indicating that the addition of PF-1240 weakened the motility of bacteria.

### 3.9. Effect of PF-1240 on Siderophore of H. alvei

Iron is essential for the formation of biofilms, which regulate surface movement and stabilize the polysaccharide matrix. The secretion of microbial siderophores is regulated by QS [44,45]. Under conditions of iron deficiency, the hydrophobicity of the microbial surface decreases, which limits the formation of biofilms [46]. The growth of different concentrations of PF-1240 in the CAS plate is shown in Figure 4C. With an increase in enzyme concentration, the orange transparent circle gradually decreases. When the enzyme dosage was 25 μg/mL, there was no orange transparent circle. Diggle et al. [47] reported that *P. aeruginosa* regulates the production of siderophores through quorum sensing. Through the analysis and determination of the relative content of siderophores, we found that with an increase in PF-1240 concentration (Figure 4D), the relative content of ferritin gradually decreased, and the maximum reduction was 54%.

### 3.10. Effect of PF-1240 on Extracellular Protease of H. alvei

Extracellular proteases are the main virulence factors of bacteria. They not only provide the necessary nutrients for bacterial growth but also directly destroy the host immune defense system. By measuring the changes in extracellular protease, we can understand the effect of PF-1240 on the expression of *H. alvei* extracellular protease. The experimental results are shown in Figure 5A. With an increase in enzyme concentration, the transparent circle gradually decreased, and when 25 μg/mL enzyme was added, the activity of extracellular protease was completely inhibited. As QS is a transcriptional regulation system, it may intervene in the expression of extracellular proteases by regulating the transcription of genes encoding extracellular proteases. Li et al. [48] found that the QS system in *P. aeruginosa* interferes with the expression of three extracellular proteases. By constructing QS mutants, it was shown that the QS system not only regulates the activity of these proteases through transcription but also controls their activity even if these proteases are secreted from cells [49]. Based on the function of PF-1240, it can be speculated that quenching AHLs can inhibit the expression of extracellular protease but do not affect its activity.

### 3.11. Effect of PF-1240 on Lipase of H. alvei

Many studies have shown that the expression of the bacterial lipase gene is dependent on cell density. So far, the molecular mechanism of QS system activation of lipase gene expression has only been explained in *P. aeruginosa*. To further determine whether PF-1240 has an effect on the lipase of *H. alvei*, we observed the effect of different concentrations of PF-1240 on the lipase in the supernatant of *H. alvei* using the plate diffusion method. It can be seen that the strain lacks a strong ability to produce lipase, but with the addition of PF-1240, the transparent circle around the hole gradually decreases (Figure 5B), indicating that the addition of PF-1240 will interfere with the production of lipase. The specific mechanism needs to be further studied.

## 4. Conclusions

We isolated and characterized a novel quorum quenching enzyme (PF-1240) with bifunctional capability to degrade a variety of AHLs, and its potential for quorum quenching was assessed. Through sequence alignment and heterologous expression, it was preliminarily inferred that it was an AHL acylase. GC-MS analysis showed that PF-1240 quenched AHLs with different acyl chain lengths. It was found that PF-1240 interfered with the QS phenomenon of *H. alvei* and inhibited the production of virulence factors. Quorum sensing is a unique regulatory mechanism in microorganisms and plays an important role in the study of food corruption caused by bacteria. It has become a potential target for human beings to regulate microbial behavior and has shown its application potential in the fields of food preservation and biological fermentation. These results indicate that PF-1240 can be used as a potential QS inhibitor to inhibit QS in aquatic spoilage bacteria. As an exogenous substance, the less it is added to food, the better. Therefore, in the next step, we will study the 3D structure of PF-1240, explore its quenching mechanism, and further improve its activity through directional evolution in order to use it to inhibit the QS of more aquatic spoilage bacteria.

## Figures and Tables

**Figure 1 foods-10-02700-f001:**
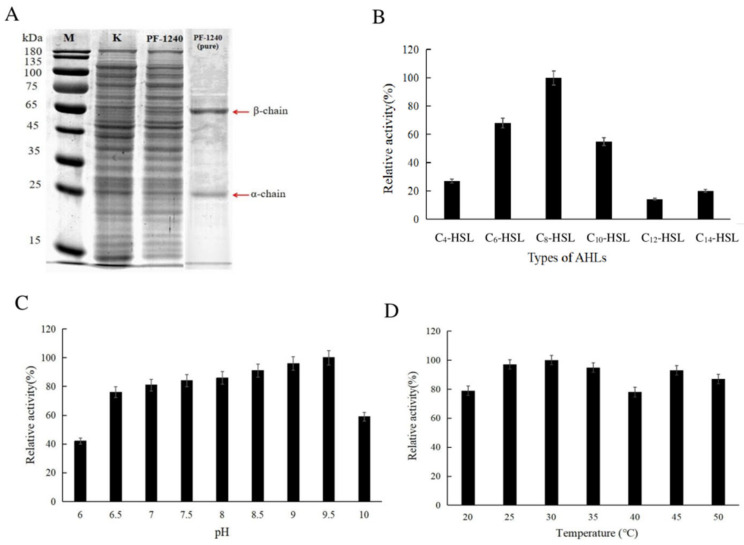
Expression and enzymatic properties of PF-1240. (**A**) SDS-PAGE analysis of purified PF-1240. Arrows indicate bands corresponding to two subunits of PF-1240. Lane M, molecular size marker; lane PF-1240, crude proteins; lane PF-1240 (pure), purified PF-1240. (**B**) Optimum reaction substrate of PF-1240. (**C**) Optimum reaction pH of PF-1240. (**D**) Optimum reaction temperature of PF-1240.

**Figure 2 foods-10-02700-f002:**
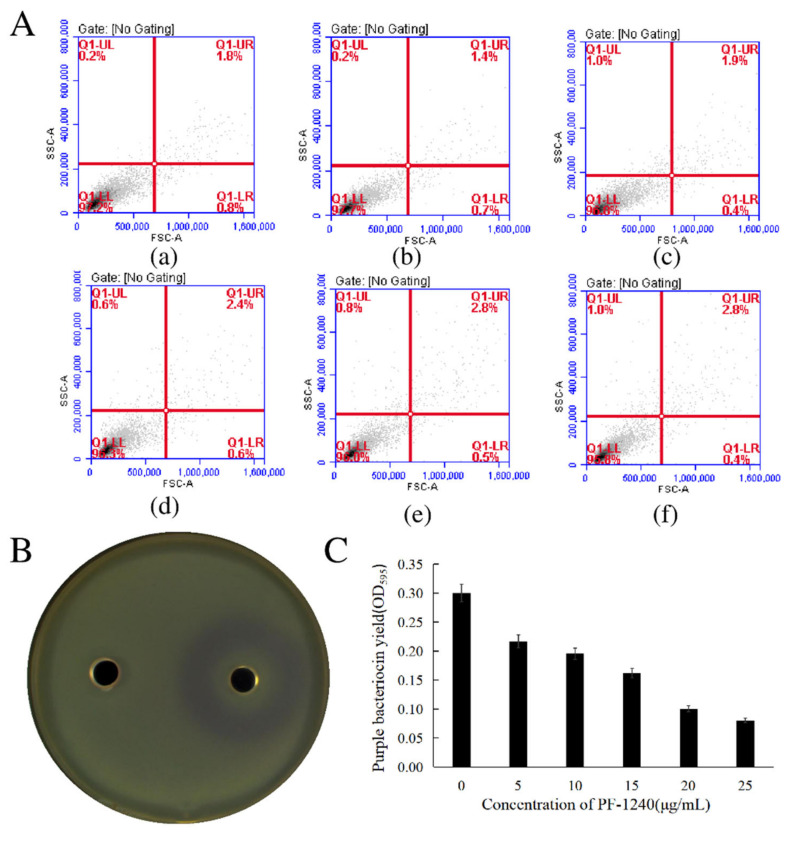
Determination of QSI properties of PF-1240. (**A**) Effects of different concentrations of PF-1240 on the growth of *H. alvei* (a–f respectively represent bacterial solution after adding 0, 5, 10, 15, 20, and 25 μg/mL of PF-1240 culture). (**B**) Inhibition of PF-1240 on CV026. (**C**) Effect of different concentrations of PF-1240 on yield of puromycin of CV026 strain.

**Figure 3 foods-10-02700-f003:**
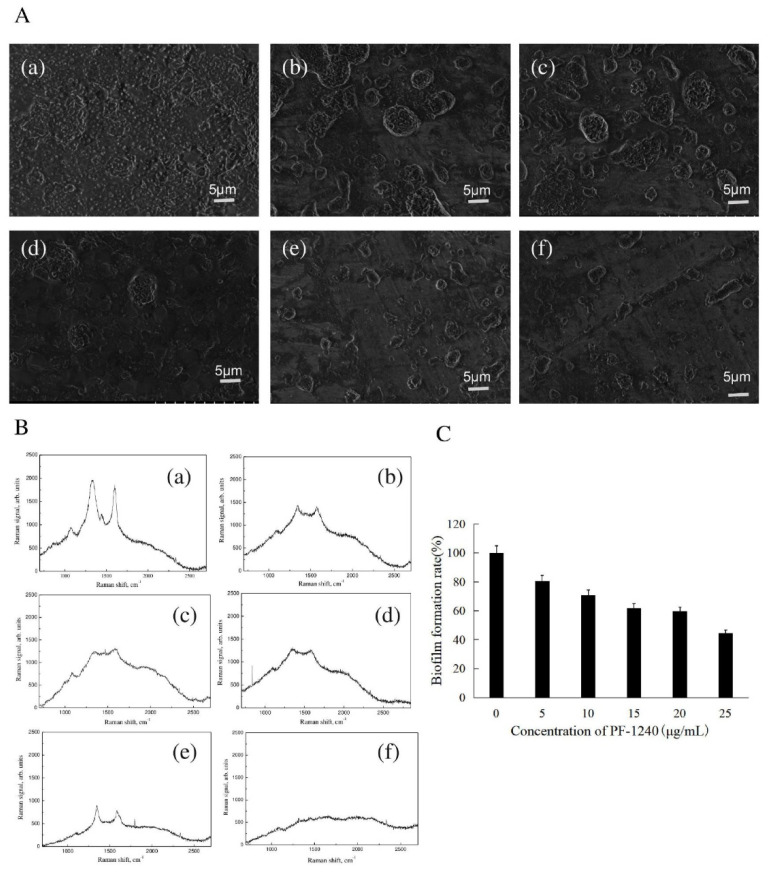
Effect of PF-1240 on biofilm and extracellular polymer formation of *H. alvei*. (**A**) SEM of biofilm cultured with different concentrations of PF-1240. (**B**) Effects of different concentrations of PF-1240 on relative formation of biofilm. (**C**) Raman spectra of *H. alvei* extracellular polymer cultured with different concentrations of PF-1240 (a–f respectively represent bacterial solution after adding 0, 5, 10, 15, 20, and 25 μg/mL of PF-1240 culture).

**Figure 4 foods-10-02700-f004:**
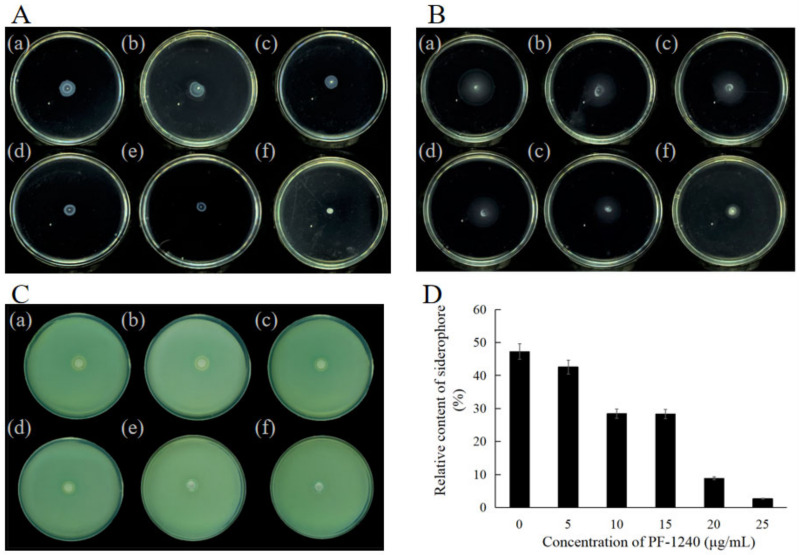
Effect of PF-1240 on motility and siderophore of *H. alvei*. (**A**,**B**) Effects of different concentrations of PF-1240 on swarming and swimming. (**C**,**D**) Effects of different concentrations of PF-1240 on relative formation of siderophore (a–f respectively represent bacterial solution after adding 0, 5, 10, 15, 20, and 25 μg/mL of PF-1240 culture).

**Figure 5 foods-10-02700-f005:**
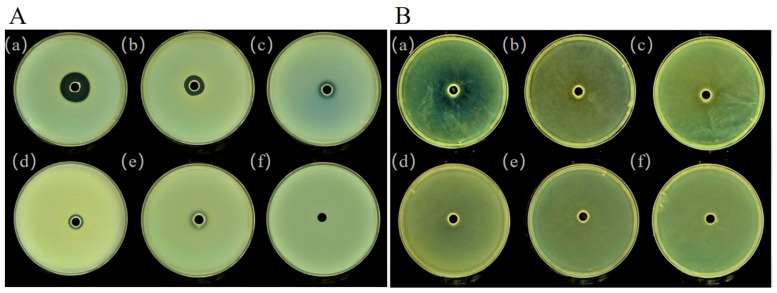
Effect of PF-1240 on extracellular protease and lipase of *H. alvei*. (**A**) Effects of different concentrations of PF-1240 on extracellular protease. (**B**) Effects of different concentrations of PF-1240 on lipase (a–f respectively represent bacterial solution after adding 0, 5, 10, 15, 20, and 25 μg/mL of PF-1240 culture).

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
