# Peer review of "Quorum Quenching Enzyme (PF-1240) Capable to Degrade AHLs as a Candidate for Inhibiting Quorum Sensing in Food Spoilage Bacterium Hafnia alvei"

_foods, 2021, doi:10.3390/foods10112700_

Round 1

Reviewer 1 Report

Thank you for submitting the manuscript entitled, “Quorum quenching enzyme (PF-1240) with degradation of AHLs as candidate for inhibiting quorum sensing Hafnia alvei in spoilage aquatic products.” Please see the comments below regarding the manuscript.

  1. Question 1, line 124 and line 188: what was the culture volume for the bacteria or what was the concentration of the bacteria (OD600)?
  2. Question 2, lines 155-156: what are the specifications of the instrument and parameters used to obtain the mass spectra?
  3. Question 3, lines 127-128: did you extract the bacterial four times with equal volumes of ethyl acetate then repeat the entire process three times? It seems unclear as to how many times the bacterial culture was extracted.
  4. Question 4, line 129: were the ethyl acetate fractions from the entire process combined then evaporated or is “mixed” indicating the ethyl acetate fractions were stirred?
  5. Question 5, Figure 4: where is a, b, c, d, e, and f on the figure? It is unclear as to where these letters can be found.

Author Response

Dear Reviewers:

Thanks very much for taking your time to review this manuscript. I really appreciate all your comments and suggestions! Please find my itemized responses in below and my revisions in the re-submitted files.

Thanks again!

Reviewer 1

Question 1, line 124 and line 188: what was the culture volume for the bacteria or what was the concentration of the bacteria (OD600)?

Response: The culture volume for the bacteria is 300 mL, we have added to line 124.The the concentration of the bacteria (OD600) is 109 CFU/mL.

Question 2, lines 155-156: what are the specifications of the instrument and parameters used to obtain the mass spectra?

Response: We have added the specifications of the instrument and parameters used to obtain the mass spectra to line 156-163.

Question 3, lines 127-128: did you extract the bacterial four times with equal volumes of ethyl acetate then repeat the entire process three times? It seems unclear as to how many times the bacterial culture was extracted.

Response: No, there are some ambiguities here. We extracted the bacterial three times with equal volumes of ethyl acetate, then collect all ethyl acetate and steam it.We have revised the expression of this part in line 128 .

Question 4, line 129: were the ethyl acetate fractions from the entire process combined then evaporated or is “mixed” indicating the ethyl acetate fractions were stirred?

Response: The ethyl acetate fractions from the entire process combined then evaporated.  We have revised the expression of this part more clearly in line 129.

Question 5, Figure 4: where is a, b, c, d, e, and f on the figure? It is unclear as to where these letters can be found.

Response: Sorry, it was misplaced when editing,now we have re-edited the Figure 4 on page 12.

Reviewer 2 Report

Authors isolated and characterized a quorum quenching enzyme with capability to partially degrade AHLs with different chain length, and which shows quorum quenching effect. In some cases, results are not sufficiently descriptive/explained, but the work is generally well performed. Some comments are listed below:

Line 16: “In this study, a potential 16 QS quenching enzyme”, please rephrase.

Please correct the name/word “gram-positive/negative” in the whole paper which should be “G”.

Line 49-51: please rephrase.

Line 59: Hafnia alvei can also grow at temperature higher than 15-20 °C, so it can’t be considered psychrophilic.

Line 59-65: redundant, please rephrase.

Line 96: “P. fluorescens 08 isolated from turbot”, please explicit turbot.

Line 259: “The following was added to…”, please clarify.

Line 346: “bacitracin”, maybe bacteriocin?

Section 2.3: “Following previous studies, we successfully cloned the target gene” however is not clear at which study the authors are referring. This should be clarified.

Section 2.6.1 and 3.3: the authors describe a hypothetical apoptotic mechanism, which is mainly a programmed cell death, in the H. alvei. According to this reviewer this terminology should be changed as it refers to a bacteria and not to a eukaryotic cell.

As regards to the optimum reaction substrate of PF-1240, please indicates how did you measured it even in the results section.

In M&M the authors describe the extraction procedure of H. alvei AHLs. However, it is not sufficiently clear (not described in the results section) how the AHLs have been detected and/or identified and/or used, as some AHL molecules have been acquired from companies.

Please revise figures/pictures as the results are not very well visible, e.g. Fig. 2B, Fig. 3A, Fig. 4, Fig. 5.

Minor spell mistakes are present, please check.

Author Response

Dear Reviewers:

Thanks very much for taking your time to review this manuscript. I really appreciate all your comments and suggestions! Please find my itemized responses in below and my revisions in the re-submitted files.

Thanks again!

Reviewer 2

Comment: Line 16: “In this study, a potential 16 QS quenching enzyme”, please rephrase.

Response: We have modified it to “In this study, we found a newl QS quenching enzyme (Line 16).

Comment: Please correct the name/word “gram-positive/negative” in the whole paper which should be “G”.

Response: We have completed the modification. (Lines 29, 48, 275)

Comment: Line 49-51: please rephrase.

Response: We have changed this sentence to “such as production of extracellular protease ......”.

Comment: Line 59: Hafnia alvei can also grow at temperature higher than 15-20 °C, so it can’t be considered psychrophilic.

Response: We agree with you, the statement here may be incorrect. We have changed the contents of it to “H. alvei is a common dominant spoilage bacterium in vacuum packaging and modified atmosphere packaging food.”.

Comment: Line 59-65: redundant, please rephrase.

Response: We have deleted the redundant part, please see lines 61-64.

Comment: Line 96: “P. fluorescens 08 isolated from turbot”, please explicit turbot.

Response: We have added a description of turbot on lines 96-97.

Comment: Line 259: “The following was added to…”, please clarify.

Response: We have modified the statement here.

Comment: Line 346: “bacitracin”, maybe bacteriocin?

Response: Yes. There is a spelling error here. We have modified it.(Line 346, page 8)

Comment: Section 2.3: “Following previous studies, we successfully cloned the target gene” however is not clear at which study the authors are referring. This should be clarified.

Response: This is our previous research. The purpose is to verify the whole genome sequencing results and clone the target gene for the construction of expression vector. We have revised the statement here more clearly. (Section 2.3, Line 111)

Comment: Section 2.6.1 and 3.3: the authors describe a hypothetical apoptotic mechanism, which is mainly a programmed cell death, in the H. alvei. According to this reviewer this terminology should be changed as it refers to a bacteria and not to a eukaryotic cell.

Response: We know what you mean. In this part, we mainly study the state of bacteria in each group after the end of culture, so as to verify whether PF-1240 has an effect on bacterial growth, rather than explore its death mechanism. We have increased the expression of this issue.(Lines 273-274)

Comment: As regards to the optimum reaction substrate of PF-1240, please indicates how did you measured it even in the results section.

Response: We have added the expression of this part.(Line 248-249)

Comment: In M&M the authors describe the extraction procedure of H. alvei AHLs. However, it is not sufficiently clear (not described in the results section) how the AHLs have been detected and/or identified and/or used, as some AHL molecules have been acquired from companies.

Response: The separation of AHLs is a complex and long process, so we used the standard from companies when determining the enzymatic properties of PF-1240. However, when determining the inhibitory effect of PF-1240 on quorum sensing, AHLs extracted from H. alvei were used. We have added a description of these.(Lines 109,135).

Comment: Please revise figures/pictures as the results are not very well visible, e.g. Fig. 2B, Fig. 3A, Fig. 4, Fig. 5.

Response: We have made changes.(Fig. 2, Fig. 3, Fig. 4, Fig. 5)

Comment: Minor spell mistakes are present, please check.

Response: We have made changes.